# Implementation of Unbalanced Ternary Logic Gates with the Combination of Spintronic Memristor and CMOS

**Haifeng Zhang [1], Zhaowei Zhang [1], Mingyu Gao [1], Li Luo [2], Shukai Duan [2], Zhekang Dong [1,3,\*] and Huipin Lin [1,3,\*]**

[1] College of electronic information, Hangzhou Dianzi University, Hangzhou 310018, Zhejiang, China; hfzhang0811@hdu.edu.cn (H.Z.); zzw0211@hdu.edu.cn (Z.Z.); mackgao@hdu.edu.cn (M.G.)

[2] School of Artificial Intelligence, Southwest University, Chongqing 400715, China; lltaozi12@email.swu.edu.cn (L.L.); duansk@swu.edu.cn (S.D.)

[3] College of Electrical Engineering, Zhejiang University, Hangzhou 310027, Zhejiang, China

\* Correspondence: englishp@hdu.edu.cn (Z.D.); linhuipin@hdu.edu.cn (H.P.)

**Abstract:** A memristor is a nanoscale electronic element that displays a threshold property, non-volatility, and variable conductivity. Its composite circuits are promising for the implementation of intelligence computation, especially for logic operations. In this paper, a flexible logic circuit composed of a spintronic memristor and complementary metal-oxide-semiconductor (CMOS) switches is proposed for the implementation of the basic unbalanced ternary logic gates, including the NAND, NOR, AND, and OR gates. Meanwhile, due to the participation of the memristor and CMOS, the proposed circuit has advantages in terms of non-volatility and load capacity. Furthermore, the input and output of the proposed logic are both constant voltages without signal degradation. All these three merits make the proposed circuit capable of realizing the cascaded logic functions. In order to demonstrate the validity and effectiveness of the entire work, series circuit simulations were carried out. The experimental results indicated that the proposed logic circuit has the potential to realize almost all basic ternary logic gates, and even some more complicated cascaded logic functions with a compact circuit construction, high efficiency, and good robustness.

**Keywords:** spintronic memristor; unbalanced ternary logic; cascaded logic; circuit simulations

## 1. Introduction

Over the past years, Moore's law seems to have stagnated as the demand for electronic devices to be scaled down has become increasingly difficult to be met [1]. The desire for new materials/mechanism-based devices that are compatible with the traditional complementary metal-oxide-semiconductor (CMOS) is realistic and attractive. The memristor, postulated by Leon Chua in 1971 [2] and physically implemented by a Hewlett Packard (HP) lab in 2008 [3], has become one of the remedies for addressing the continued scaling down of modern electronic circuits. As a passive nanoscale component, a memristor possesses many superior properties, including the nonvolatility, high density, continuous input/output property, threshold property, and variable conductivity [4–6]. All these above-mentioned advantages make the memristive device a powerful candidate in intelligent computation [7–10], with logic operation as an example [10].

So far, memristor-based logic implementation has been gaining considerable attention and many different design approaches have been successively developed [11–14]. Almost all the existing work mainly focuses on the investigation of binary logic. The work related to the memristor-based multi-value logic (MVL) implementation is relatively rare and incomplete. Compared with the binary

logic, MVL provides exponentially higher data density with lower circuit and interconnect overheads, leading to lower parasitic effects, power consumption, and delays. As the simplest form of the MVL, the ternary logic was the first to be investigated in this study, which is in accordance with the basic scientific research rule (i.e., from simple to difficult). Notably, based on the external voltage, ternary logic can be further divided into two categories, i.e., the unbalanced ternary logic and the balanced ternary logic [15,16]. In the former one (i.e., the unbalanced ternary logic), only positive voltages are used, namely {0, 1, 2}; in the latter one (i.e., the balanced logic), both the positive and negative voltages are used as {−1, 0, 1} [15,16]. In 1984, the first implementation of ternary logic gates was provided based on the mature CMOS technology, which could perform the functions of basic ternary gates [17]. However, the entire circuit was highly sensitive to the transistor dimensions, and the robustness could not be sufficiently guaranteed. Then, Lin et al. [18] aimed to implement the convenient ternary logic gates based on carbon nanotube field-effect transistors (CNTFETs). However, the CNTFETs-based ternary logic suffers from the "charge pile-up" issue in the channel that may affect the performance of on/off switching [19]. Considering the multi-states ability of the memristors, the ternary logic gates have recently been realized by memristive devices [12,16,20]. Khalid and Singh [16] used the simplest HP memristor model, which is not practical and cannot represent the real device characteristics. In other words, more realistic models should be used for the implementation of ternary logic. Hence, this work utilized the spintronic memristor [21–23], which can reflect realistic device properties and has been proved in many memristor-based applications [24]. Soliman et al. [20] can perform basic ternary logic operations by using the memristor and CNTFET, while the functionality is limited and the robustness is not good enough. Based on these, this study further investigated the memristor-based unbalanced ternary logic implementation, and the main contributions can be summarized as below:

- A flexible memristor-CMOS-based logic circuit was designed, which could perform the different ternary logic gates (i.e., the NAND, NOR, AND, and OR gates) by easily changing the polarities of the interconnected memristors.
- Due to the participation of the spintronic memristor, the logic states can be stored in the memristor. That is, the logic computation and data storage are integrated into the proposed logic circuit, which opens up a new path to explore the new intelligent computation systems, in contrast to the classical von Neumann system with separated computation and storage configuration.
- The CMOS switches installed in the reading circuit means the entire logic circuit has a sufficient load capacity, and the signal degradation issue can also be addressed effectively.

The rest of the paper is organized as below. The spintronic memristor with its resistance variation rule is discussed in Section 2. Based on this, a flexible memristor-CMOS logic circuit, which can perform four basic ternary logic gates, is presented in Section 3. Meanwhile, a brief comparison and analysis of six different ternary logic implementations are carried out in the same section. For verification, a series of circuit simulations (two case studies) with the relevant analysis are conducted in Section 4. Finally, Section 5 concludes the paper.

## 2. Spintronic Memristor

Different from the solid-state memristor [2,3], magnetic technology provides other possibilities to build up a memristive system. Wang et al. [21] proposed three possible physical structures of a spintronic memristor. The memristive effect can be realized using the spin-torque-induced magnetization switching or the magnetic-domain-wall motion. Compared to a solid-state thin-film device [2,3], the electrical behavior (describing the relationship between the memristance and the current passing through the memristor) can be controlled more flexibly. Meanwhile, among all the spintronic memristor models, the spin-valve memristor with magnetic-domain-wall motion could be the most suitable candidate due to its compact and simple structure [22,23]. Its three-dimensional (3D) structure and corresponding simplified circuit model is shown in Figure 1.

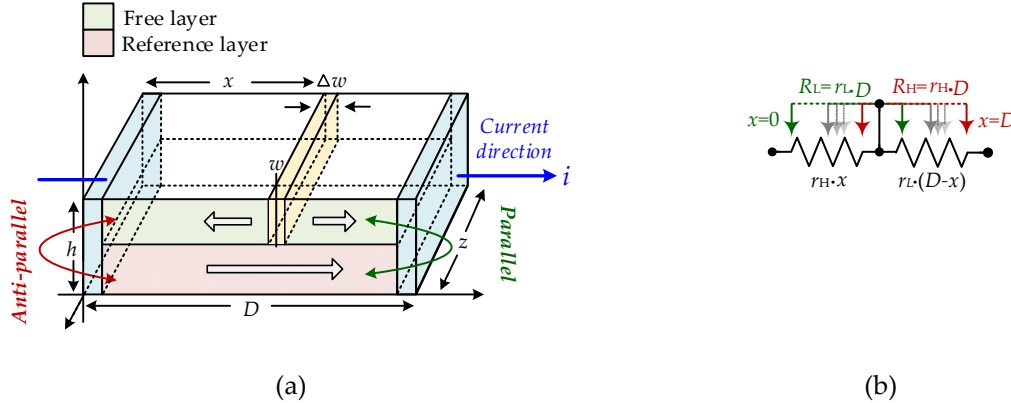

(a)           (b)

**Figure 1.** (**a**) 3D structure of a current in-plane spintronic memristor and (**b**) a simplified equivalent circuit.

Figure 1a illustrates the basic structure and physical principle of a current in-plane (CIP) spintronic memristor. This device can be considered a long spin-value strip with the size of $D$ (length) $\times z$ (width) $\times h$ (height). It consists of two up-down ferromagnetic layers called the free layer and reference layer, respectively. The free layer is divided by a domain-wall into two segments with opposite magnetization directions, where the domain wall motion is driven by spin-polarized current and the scattering of spin-polarized current depends on the position of the domain wall [25]; the reference layer is an integral whole with a fixed magnetization direction. Notably, resistance dependence on the relative orientation of two coupled magnetic layers is called giant magnetoresistance (GMR) and was described theoretically by Camley and Barnaś [26]. Specifically, the resistance per unit length of each segment completely depends on the relative magnetic directions of the free layer and the reference layer. Specifically, when the magnetic direction of the free layer is parallel (anti-parallel) to the reference layer, the resistance per unit length is low (high).

The mathematical expression for the resistance of a spintronic memristor can be written as [23]:

$$M(x) = r_H{\cdot}x + r_L{\cdot}(D - x) \tag{1}$$

where $r_H$ and $r_L$ denote the highest and lowest resistance per unit length, respectively. The variable $x$ represents the position of the domain wall, and its dynamic function is given as [23]:

$$v = \frac{dx}{dt} = \begin{cases} \eta{\cdot}\Gamma v{\cdot}J, & J \geq J_{cr} \\ 0, & J < J_{cr} \end{cases} \tag{2}$$

where $\eta = \pm 1$ denotes the polarity of the spintronic memristor. $\Gamma_v$ is the domain wall velocity coefficient, and $J$ and $J_{cr}$ are the real-time and critical current density, respectively. Then, the critical current $I_{cr}$ can be calculated using $I_{cr} = J_{cr}{\cdot}h{\cdot}z$. The domain wall movement occurs only when the real-time current density $J$ is above the critical current density $J_{cr}$.

Differentiating Equation (1) with respect to $t$ gives the memristance variation rate as follows:

$$\frac{dM}{dt} = \Delta r{\cdot}\frac{dx}{dt} = \begin{cases} \Delta r{\cdot}\eta{\cdot}\Gamma_v{\cdot}J, & J \geq J_{cr} \\ 0, & J < J_{cr} \end{cases} \tag{3}$$

where $\Delta r$ denotes the difference between $r_H$ and $r_L$, i.e., $\Delta r = r_H - r_L$.

From Equation (3), if the real-time current density $J$ is smaller than the critical current density $J_{cr}$ (namely, $J < J_{cr}$), the memristance variation is equal to zero, the spintronic memristor can be deemed an ordinary resistor; otherwise (i.e., $J \geq J_{cr}$), Equation (3) can be rewritten as:

$$\frac{dM}{dt} = \Delta r{\cdot}\eta{\cdot}\Gamma_v{\cdot}J = \frac{\Delta r{\cdot}\eta{\cdot}\Gamma_v}{h{\cdot}z}{\cdot}\frac{V}{M} \tag{4}$$

where $V$ is the voltage applied to the spintronic memristor.

By integrating both sides of Equation (4), Equation (1) can be rewritten as:

$$M(\varphi) = \begin{cases} R_H, & \varphi > \varphi_{th2} \\ \sqrt{M_0^2 + 2A\varphi}, & \varphi_{th1} \leq \varphi \leq \varphi_{th2} \\ R_L, & \varphi < \varphi_{th1} \end{cases}, \text{where} \begin{cases} \varphi_{th1} = \frac{R_L^2 - M_0^2}{2A} \\ \varphi_{th2} = \frac{R_H^2 - M_0^2}{2A} \end{cases} \quad (5)$$

where $\varphi$ is the magnetic flux flowing through the spintronic memristor and the flux thresholds ($\varphi_{th1}$ and $\varphi_{th2}$) are determined by the natural memristance boundary: $R_H = r_H \cdot D$ and $R_L = r_L \cdot D$ (commonly $R_H \gg R_L$). $M_0$ denotes the initial memristance at $t = 0$, and $A = \Delta r \cdot \eta \cdot \Gamma_v / h / z$ is an auxiliary constant.

From Equation (5), if the real-time current density $J$ always satisfies $J \geq J_{cr}$, the total flux variation $\Delta\varphi$ can be calculated using:

$$\Delta\varphi = \frac{1}{2A}\left[M_{Obj}^2 - M_0^2\right] \quad (6)$$

where $M_{obj}$ denotes the target memristor. Furthermore, assuming the applied voltage is a constant $V_{con}$, the switching time $\Delta T$ can be computed using $\Delta T = \Delta\varphi/V_{con}$.

To facilitate the circuit simulation in the subsequent sections, the corresponding Spice model of a spintronic memristor was built up. The relevant sub-circuit description is provided in Table 1.

**Table 1.** Spice model description for spintronic memristor.

| *Spintronic Memristor |
| --- |
| .SUBCKT Spintronic memristor Plus Minus Flux Charge PARAMS: |
| +D=1000E-9 h=70E-10 z=10E-9 rl=4E9 rh=6E9 Jcr=5E11 Taov=1.3517E-11 |
| ******* Differential equation modeling****** |
| Gx 0 x value={if(abs(I(Emem)/(h*z))<Jcr,0,Taov*I(Emem)/(h*z))} |
| Cx x 0 1 IC={0} |
| Raux x 0 1T |
| ******* Resistive port of the memristor****** |
| Emem plus aux value={if(V(x)<=D, I(Emem)*V(x)*(rh-rl), I(Emem)*D*(rh-rl))} |
| RH aux minus {rl*D}*******Flux computation ****** |
| Eflux flux 0 value={SDT(V(plus, minus))} |
| *******Charge computation ****** |
| Echarge charge 0 value={SDT(I(Emem))} |
| .ENDS spintronic memristor |

Notably, the memristance variation rule of the spintronic memristor, which is very important for the implementation of memristor-based multi-valued logic, is discussed through the use of PSpice circuit simulations (as shown in Figure 2, Version 16.5, Cadence, San Jose, CA, USA).

Figure 2a illustrates the memristance variation under a positive voltage pulse $V+$ (Amplitude: (0.15, 0.16, 0.18, 0.2, 0.22) for each period = 80 ns). $V_{th1} = R_L \cdot I_{cr}$ and $V_{th2} = R_H \cdot I_{cr}$ were the voltage thresholds, and the initial memristance was set to $M_0 = R_L$. In each period, the memristance variation could be divided into two phases as follows:

Phase 1: The real-time current density satisfied $J \geq J_{cr}$, where the memristance increased while the real-time current density decreased.

Phase 2: Until the real-time current density satisfied $J < J_{cr}$, the memristance tended to be steady.

Figure 2b illustrates the memristance variation under a negative voltage pulse $V-$. Similarly, the memristance variation could be divided into two phases.

Phase 1: When $0 \leq t \leq 200$ ns, the real-time current density satisfied $J < J_{cr}$ and the memristance remained at the initial value $R_H$.

Phase 2: Once the real-time current density satisfied $J \geq J_{cr}$ (i.e., $200$ ns $\leq t \leq 400$ ns), the memristance decreased to its lowest value $R_L$ within a short time.

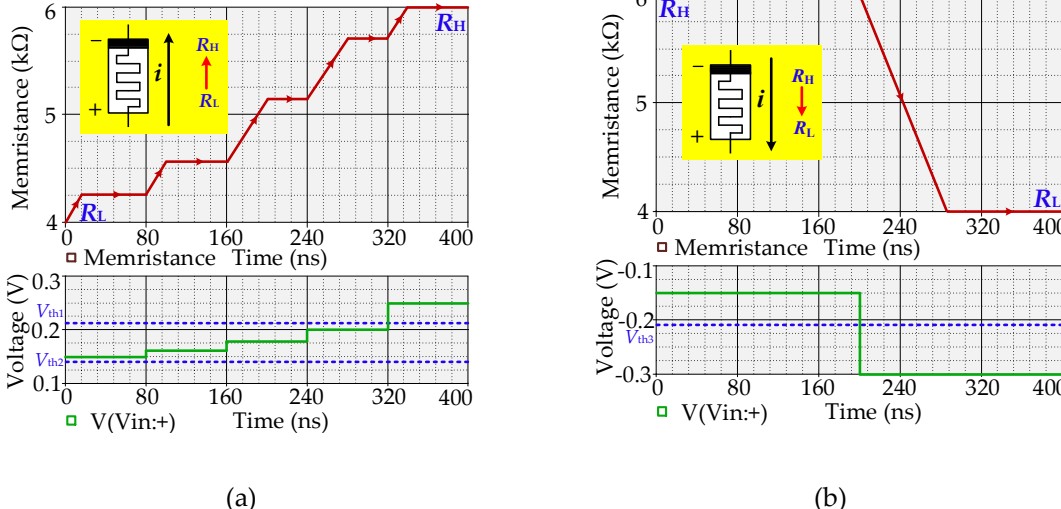

**Figure 2.** (**a**) The memristance variation under a positive voltage pulse. (**b**) The memristance variation under a negative voltage pulse.

## 3. Implementation of Ternary Logic Gates

In this section, a specific description of the design process of a flexible memristor-CMOS hybrid circuit for the implementation of ternary logic gates (NAND, NOR, AND, and OR gates) is given.

### 3.1. Memristor-CMOS Hybrid Circuit

The memristor-CMOS hybrid circuit diagram for the implementation of ternary logic gates is provided in Figure 3. Note that the entire circuit can be divided into three parts: the leftmost box is the so-called initialization circuit, the middle box is the writing circuit, and the rightmost box provides the corresponding reading circuit.

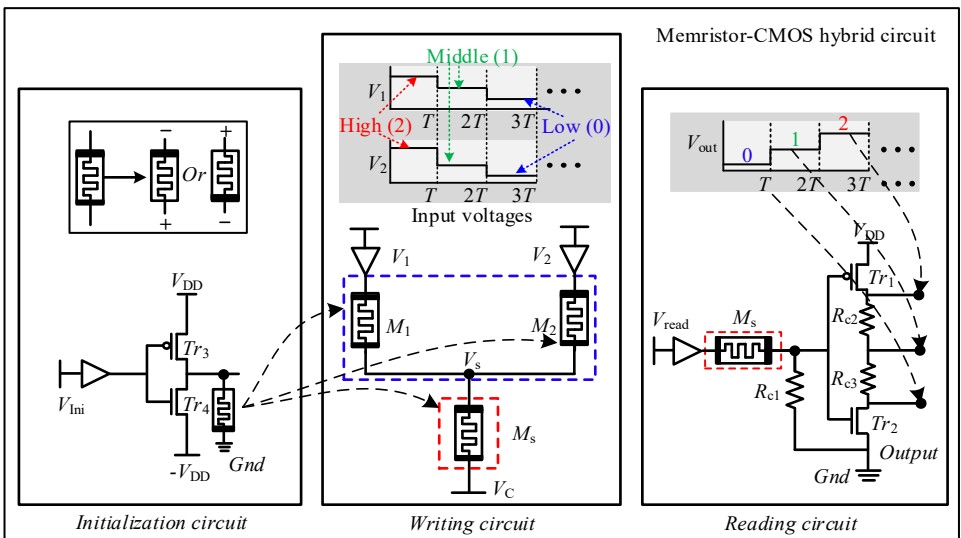

**Figure 3.** The memristor-CMOS (complementary metal-oxide semiconductor) hybrid logic circuit (including initialization circuit, writing circuit, and reading circuit).

In the writing circuit (as shown in Figure 3), $M_1$ and $M_2$ (labeled by the blue dashed box) are two identical spintronic memristors with the boundary resistances $R_L$ and $R_H$. Namely, their resistances ($R_1$ and $R_2$) satisfy $R_{1,2} \in [R_L, R_H]$. The remaining memristor $M_s$ (labeled by the red dashed box) is named the load memristor. Its resistance ($R_s$) varies from $R_{SL}$ to $R_{SH}$, i.e., $R_s \in [R_{SL}, R_{SH}]$. Notably, to realize

different ternary logic gates, the three resistances should always satisfy $R_s \gg R_{1,2}$. $V_1$ and $V_2$ are two time-sequence inputs with the period $T$, and $V_c$ is a constant voltage. In the initialization circuit, $V_{ini} = \pm V_{DD}$ denotes the initialization voltage for the resistance initialization. In the reading circuit, $V_{read}$ represents the reading voltage. Its value satisfies $V_{read} < R_{SL} \cdot I_{cr}$ such that the load memristance variation will not occur during the reading operation. $R_{c1}$, $R_{c2}$, and $R_{c3}$ are three regular resistors.

Then, it can be concluded that a ternary logic operation can be realized using the proposed memristor-CMOS hybrid circuit within three steps, namely initialization, writing operation, and reading operation. The input voltages ($V_1$ and $V_2$) and the output voltage $V_{out}$ represent the input and output logic state variations, respectively. The specific realization process is provided in the next subsection.

*3.2. NAND Gate*

Theoretically, the NAND gate can be deemed a universal gate. Namely, every other gate function can be generated by successive implementations of NAND gates [10]. Hence, the specific realization of a ternary NAND gate is demonstrated below.

From Figure 3, the specific circuit diagram for the implementation of a ternary NAND gate is demonstrated in Figure 4. Based on the previous description, the entire operation is performed in three steps.

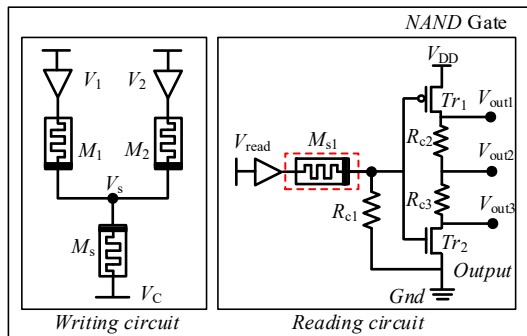

**Figure 4.** The circuit diagram for the implementation of a NAND gate (including writing circuit and reading circuit).

Step 1: Initialization

As the name suggests, the initialization is conducted by the initialization circuit (as shown in Figure 3). Based on the memristance variation rule, when the initialization voltage $V_{ini} = V_{DD}$, the upper transistor $Tr_3$ turns off, while the lower transistor $Tr_4$ turns on. In this situation, the voltage applied to the memristor is equal to $-V_{DD}$ and the resistance of the memristor will decrease. On the contrary, when the initialization voltage $V_{ini} = -V_{DD}$, the transistor $Tr_3$ turns on and the lower transistor $Tr_3$ turns off. The voltage applied to the memristor is equal to $V_{DD}$ and the memristance will increase. Before each ternary NAND operation, the memristors $M_1$ and $M_2$ need to initialize to their lowest value $R_L$, and it is recommended that the memristor $M_{s1}$ is initialized to a median value $R_{Mid}$.

Step 2: Writing operation

The writing operation is performed by the writing circuit. Based on Kirchhoff's current law (KCL), the current flowing in the writing circuit can be written as:

$$\frac{V_1 - V_s}{R_1} + \frac{V_2 - V_s}{R_2} = \frac{V_s - V_c}{R_{s1}} \tag{7}$$

Due to $R_{s1} \gg R_{1,2}$, the node voltage $V_s$ can be approximately calculated using:

$$V_s \approx \frac{R_2}{R_1 + R_2} \cdot V_1 + \frac{R_1}{R_1 + R_2} \cdot V_2 \tag{8}$$

For ternary logic operations, the inputs ($V_1$ and $V_2$) always have three states $V_H$, $V_L$, and $V_{Mid}$, representing the logic "2," logic "1," and logic "0," respectively. Then, the six possible cases can be summarized as follows:

- Case a: When $V_1 = V_2 = V_H$ (logic "2"), the node voltage $V_s \approx V_H$ (From Equation (8)). The voltage applied to the load memristor $M_{s1}$ is equal to $V_H − V_c$. Assuming $V_H − V_c > R_{Mid} \cdot I_{cr}$, the load memristance $R_{s1}$ will go down to its lowest value $R_{SL}$ within a very short time.

- Case b: When $V_1 = V_2 = V_{Mid}$ (logic "1"), the node voltage $V_s \approx V_{Mid}$. The voltage applied to the load memristor $M_{s1}$ is equal to $V_{Mid} − V_c$. Assuming $|V_{Mid} − V_c| \leq R_{Mid} \cdot I_{cr}$, the load memristance $R_{s1}$ remains in the initial state $R_{Mid}$.

- Case c: When $V_1 = V_2 = V_L$ (logic "0"), the node voltage $V_s \approx V_L$. The voltage applied to the load memristor $M_{s1}$ is equal to $V_c − V_L$. Assuming $|V_c − V_L| > R_{Mid} \cdot I_{cr}$, the load memristance $R_{s1}$ will sharply increase to its highest value $R_{SH}$.

- Case d: When $V_1 = V_L$ and $V_2 = V_H$ (or $V_1 = V_H$ and $V_2 = V_L$), the node voltage $V_s$ can be calculated using:

$$V_s \approx \begin{cases} \frac{R_1}{R_1 + R_2} \cdot V_H \text{ (if } V_1 = V_L \text{ and } V_2 = V_H) \\ \frac{R_2}{R_1 + R_2} \cdot V_H \text{ (if } V_1 = V_H \text{ and } V_2 = V_L) \end{cases} \tag{9}$$

According to the memristance variation rule, the node voltage $V_s$ will vary to $R_L/(R_L+R_H) \cdot V_H \approx 0$ (due to $R_H \gg R_L$). Similar to Case c, the load voltage is equal to $V_c − V_L$, and the load memristance $R_{s1}$ will go up to its highest value $R_{SH}$.

- Case e: When $V_1 = V_L$ and $V_2 = V_{Mid}$ (or $V_1 = V_{Mid}$ and $V_2 = V_L$), the node voltage $V_s$ can be calculated using:

$$V_s \approx \begin{cases} \frac{R_1}{R_1 + R_2} \cdot V_{Mid} \text{ ( if } V_1 = V_L \text{ and } V_2 = V_{Mid}) \\ \frac{R_2}{R_1 + R_2} \cdot V_{Mid} \text{ (if } V_1 = V_{Mid} \text{ and } V_2 = V_L) \end{cases} \tag{10}$$

Here, the voltage $V_{Mid}$ satisfies $|V_{Mid}| > R_L \cdot I_{cr}$, the node voltage $V_s$ will change to $R_L/(R_L+R_H) \cdot V_{Mid} \approx 0$. The load memristance will increase to the highest value $R_{SH}$.

- Case f: When $V_1 = V_H$ and $V_2 = V_{Mid}$ (or $V_1 = V_{Mid}$ and $V_2 = V_H$), the node voltage $V_s$ can be expressed as:

$$V_s \approx \begin{cases} \frac{R_1}{R_1 + R_2} \cdot V_{Mid} + \frac{R_2}{R_1 + R_2} \cdot V_H \text{ ( if } V_1 = V_H \text{ and } V_2 = V_{Mid}) \\ \frac{R_2}{R_1 + R_2} \cdot V_{Mid} + \frac{R_1}{R_1 + R_2} \cdot V_H \text{ (if } V_1 = V_{Mid} \text{ and } V_2 = V_H) \end{cases} \tag{11}$$

Here, once the voltage $V_{Mid}$ satisfies $|V_H − V_{Mid}| > R_L \cdot I_{cr}$, the node voltage $V_s$ will change to $R_H/(R_L+R_H) \cdot V_{Mid} \approx V_{Mid}$. Similar to Case b, the load memristance remains in its initial state, i.e., $R_{Mid}$.

Step 3: Reading operation

After writing operation, the reading operation is performed by the reading circuit. Based on the specific load memristance, three cases can be distinguished as follows.

- Case I: When the load memristance $R_{s1} = R_{SH}$, the node voltage $V_g$ can be calculated using:

$$V_g = \frac{R_{c1}}{R_{SH} + R_{c1}} \cdot V_{read} \tag{12}$$

Assuming $R_{SH} \gg R_{c1}$, the node voltage $V_g \approx 0$. At this time, the transistor $Tr_1$ turns on, while the other transistor $Tr_2$ turns off. As a result, the output $V_{out} = V_{out1} = V_{DD} = V_H$, denoting the logic "2."

- Case II: When the load memristance $R_{s1} = R_{Mid}$, the node voltage $V_g$ can be given as:

$$V_g = \frac{R_{c1}}{R_{Mid} + R_{c1}} \cdot V_{read} \tag{13}$$

Assuming $R_{Mid} = R_{c1}$ and $R_{c2} = R_{c3}$, the node voltage is equal to $0.5V_{read}$. At this time, transistors $Tr_1$ and $Tr_2$ both turn on, and the current flows through two resistors $R_{c2}$ and $R_{c3}$. Correspondingly, the output voltage $V_{out} = V_{out2} = 0.5V_{DD} = V_{Mid}$, which denotes the logic "1."

- Case III: When the load memristance $R_{s1} = R_{SL}$, The node voltage can be computed using:

$$V_g = \frac{R_{c1}}{R_{SL} + R_{c1}} \cdot V_{read} \tag{14}$$

Assuming $R_{c1} \gg R_{SL}$, the node voltage $V_g \approx V_{read}$. Contrary to Case I, the transistor $Tr_1$ turns off, while the other transistor $Tr_2$ turns on. The output $V_{out} = V_{out3} = V_{Gnd} = V_L$, representing the logic "0."

For the purpose of clarity, a summary of the information regarding the ternary NAND gate is collected in Table 2. It is clear that the input–output relationship of the presented logic circuit is consistent with the truth table of the NAND gate, which verifies the validity of the entire operation process.

**Table 2.** A summary of the information of the ternary NAND gate.

| Truth Table [1] | | | Writing Operation | | | | Reading Operation | | |
|---|---|---|---|---|---|---|---|---|---|
| $In_1$ | $In_2$ | Out | Cases | $V_1$ | $V_2$ | $R_{s1}$ | Cases | $V_{out}$ | Logic |
| 0 | 0 | 2 | Case c | $V_L$ | $V_L$ | $R_{SH}$ | Case I | $V_H$ | 2 |
| 1 | 0 | 2 | Case e | $V_{Mid}$ | $V_L$ | $R_{SH}$ | Case I | $V_H$ | 2 |
| 2 | 0 | 2 | Case d | $V_H$ | $V_L$ | $R_{SH}$ | Case I | $V_H$ | 2 |
| 0 | 1 | 2 | Case e | $V_L$ | $V_{Mid}$ | $R_{SH}$ | Case I | $V_H$ | 2 |
| 1 | 1 | 1 | Case b | $V_{Mid}$ | $V_{Mid}$ | $R_{Mid}$ | Case II | $V_{Mid}$ | 1 |
| 2 | 1 | 1 | Case f | $V_H$ | $V_{Mid}$ | $R_{Mid}$ | Case II | $V_{Mid}$ | 1 |
| 0 | 2 | 2 | Case d | $V_L$ | $V_H$ | $R_{SH}$ | Case I | $V_H$ | 2 |
| 1 | 2 | 1 | Case f | $V_{Mid}$ | $V_H$ | $R_{Mid}$ | Case II | $V_{Mid}$ | 1 |
| 2 | 2 | 0 | Case a | $V_H$ | $V_H$ | $R_{SL}$ | Case III | $V_L$ | 0 |

[1] $In_{1,2}$ and Out denote the inputs and output of the NAND gate, respectively.

From Table 2, since both the input and output variables are represented as constant voltages, the signal degradation can be addressed in the presented logic circuit. Meanwhile, due to the existence of the CMOS in the reading circuit, the proposed logic circuit possesses a sufficient load capacity. These two advantages enable the ternary NAND gate circuit to be easily cascaded for the realization of some more complex ternary logic operations. In addition, the proposed double-input ternary NAND logic gate circuit can be further extended to realize the multi-input ternary NAND gate. The corresponding process description will not be repeated here due to the similarity with the double-input ternary NAND logic gate.

### 3.3. Other Ternary Logic Gates

Similarly, the proposed memristor-CMOS hybrid circuit can perform some other ternary logic gates, such as the NOR, AND, and OR gates. The corresponding circuit diagrams are exhibited in Figure 5. It is noted that, since the initialization circuit and reading circuit for all the above-mentioned ternary logic gates are the same (as shown in Figure 3), they are not provided in this part.

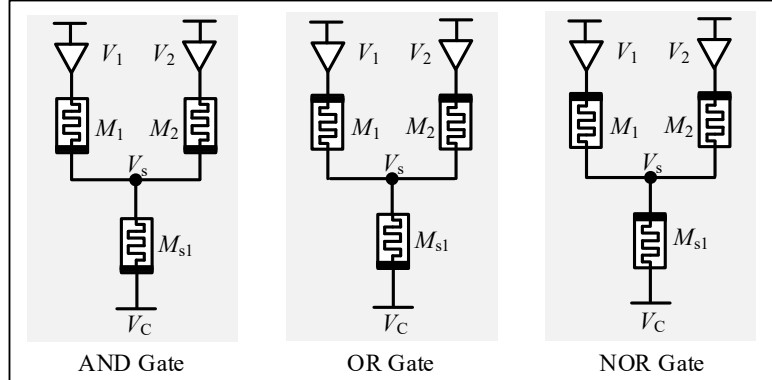

**Figure 5.** The writing circuit for the implementation of other ternary logic gates.

From Figure 5, these three ternary logic gates can be realized using the uniform circuit structure (i.e., the proposed memristor-CMOS hybrid circuit). Then, the operation steps (i.e., the initialization, writing operation, and reading operation) of these three ternary logic gates are the same as that of the two-input ternary NAND gate. In particular, during the initialization, the memristors $M_1$ and $M_2$ are both set to their highest state $R_H$, and the load memristor $R_{s1}$ is initialized to a median value $R_{Mid}$. Furthermore, due to the same circuit structure, the KCL function of all three logic circuits (AND, OR, and NOR gates) can also be mathematically expressed using Equation (7) and Equation (8). Therefore, the specific process description of these three ternary logic gates is not provided in this part due to the similarity with the two-input ternary NAND gate.

Notably, the only difference among these logic circuits (NAND, AND, OR, and NOR gates) is the polarity of the memristor (i.e., the connection mode). Hence, Figure 5 provides the important polarity information of the interconnected memristors (or the memristor connection mode), which is necessary for the implementation of different basic ternary logic gates.

For simplicity, the overall information of these logic gates is collected in Table 3.

**Table 3.** The overall information of the other ternary gates: AND, OR, and NOR gates.

| Logic Functions | Truth Table | | Initialization | | Writing Operation | | | Reading Operation | |
|---|---|---|---|---|---|---|---|---|---|
| | Input | Output | $M_1$ ($M_2$) | $M_{s1}$ | $V_1$ | $V_2$ | $R_{s1}$ | $V_{out}$ | Logic [1] |
| AND | 0 0 | 0 | $R_H$ | $R_{Mid}$ | $V_L$ | $V_L$ | $R_{SL}$ | $V_L$ | 0 |
| | 1 0 | 0 | $R_H$ | $R_{Mid}$ | $V_{Mid}$ | $V_L$ | $R_{SL}$ | $V_L$ | 0 |
| | 2 0 | 0 | $R_H$ | $R_{Mid}$ | $V_H$ | $V_L$ | $R_{SL}$ | $V_L$ | 0 |
| | 1 1 | 1 | $R_H$ | $R_{Mid}$ | $V_{Mid}$ | $V_{Mid}$ | $R_{Mid}$ | $V_{Mid}$ | 1 |
| | 2 1 | 1 | $R_H$ | $R_{Mid}$ | $V_H$ | $V_{Mid}$ | $R_{Mid}$ | $V_{Mid}$ | 1 |
| | 2 2 | 2 | $R_H$ | $R_{Mid}$ | $V_H$ | $V_H$ | $R_{SH}$ | $V_H$ | 2 |
| OR | 0 0 | 0 | $R_H$ | $R_{Mid}$ | $V_L$ | $V_L$ | $R_{SL}$ | $V_L$ | 0 |
| | 1 0 | 1 | $R_H$ | $R_{Mid}$ | $V_{Mid}$ | $V_L$ | $R_{Mid}$ | $V_{Mid}$ | 1 |
| | 2 0 | 2 | $R_H$ | $R_{Mid}$ | $V_H$ | $V_L$ | $R_{SH}$ | $V_H$ | 2 |
| | 1 1 | 1 | $R_H$ | $R_{Mid}$ | $V_{Mid}$ | $V_{Mid}$ | $R_{Mid}$ | $V_{Mid}$ | 1 |
| | 2 1 | 2 | $R_H$ | $R_{Mid}$ | $V_H$ | $V_{Mid}$ | $R_{SH}$ | $V_H$ | 2 |
| | 2 2 | 2 | $R_H$ | $R_{Mid}$ | $V_H$ | $V_H$ | $R_{SH}$ | $V_H$ | 2 |
| NOR | 0 0 | 2 | $R_H$ | $R_{Mid}$ | $V_L$ | $V_L$ | $R_{SH}$ | $V_H$ | 2 |
| | 1 0 | 1 | $R_H$ | $R_{Mid}$ | $V_{Mid}$ | $V_L$ | $R_{Mid}$ | $V_{Mid}$ | 1 |
| | 2 0 | 0 | $R_H$ | $R_{Mid}$ | $V_H$ | $V_L$ | $R_{SL}$ | $V_L$ | 0 |
| | 1 1 | 1 | $R_H$ | $R_{Mid}$ | $V_{Mid}$ | $V_{Mid}$ | $R_{Mid}$ | $V_{Mid}$ | 1 |
| | 2 1 | 0 | $R_H$ | $R_{Mid}$ | $V_H$ | $V_{Mid}$ | $R_{SL}$ | $V_L$ | 0 |
| | 2 2 | 0 | $R_H$ | $R_{Mid}$ | $V_H$ | $V_H$ | $R_{SL}$ | $V_L$ | 0 |

[1] $V_L$, $V_{Mid}$, and $V_H$ represent the logic "0," logic "1," and logic "2," respectively.

From Table 3, the input and output of these three logic circuits are all constant voltages and their relationships are all consistent with the corresponding truth table, which verifies the validity of the entire scheme.

### 3.4. Comparison and Analysis

In this subsection, five existing ternary logic implementations (i.e., the CMOS based logic [17], CNTFET based logic [18], the memristor-CNTFET based logic [20], pure memristor-based logic [16], and memristor-as-driver (MAD) logic [12]) are introduced for comparison purposes. The corresponding information (including circuit construction, input and output mode, load capacity, cascaded capacity, robustness, and functionality) is collected in Table 4.

**Table 4.** Comparison of the proposed ternary logic circuit with other logic implementation.

| Items | Method 1 [1] | Method 2 | Method 3 | Method 4 | Method 5 | Method 6 |
|---|---|---|---|---|---|---|
| Circuit Structure | Fixed | Unfixed | Unfixed | Unfixed | Unfixed | Unfixed |
| Input and Output | Voltage | Voltage | Voltage | Voltage | Memristance | Memristance |
| Load Capacity | Sufficient | Sufficient | Sufficient | Sufficient | Insufficient | Insufficient |
| Cascaded Capacity | Good | Good | Good | Good | Not good | Not good |
| Initialization | Needed | Unneeded | Unneeded | Needed | Needed | Needed |
| Robustness | Strong | Median | Median | Median | Strong | Strong |
| Functionality | Strong | Median | Median | Strong | Strong | Strong |

[1] Methods 1–6 are the proposed method, the CMOS based method, the carbon nanotube field-effect transistor (CNTFET)-based method, memristor-CNTFET based method, pure memristor-based method, and memristor-as-driver (MAD) method, respectively.

From Table 4, the circuit structure of the proposed method (i.e., method 1) is fixed and uniform compared with the other five competitors. That is, for the existing ternary logic implementation, different logic gates always need different circuit diagrams, which may lead to additional fabrication costs. Then, different from the pure memristor-based method (i.e., method 5) and the MAD method (i.e., method 6), the input and output logic state variables of the proposed method are both constant voltages, which is beneficial for addressing the signal degradation issue. Meanwhile, due to the participation of the CMOS switches, the proposed ternary logic circuit has a sufficient load capacity. Notably, based on the above two advantages, the proposed ternary logic can be used for the implementation of some more complicated logic functions with the cascaded configuration (i.e., the easily-cascaded feature). Furthermore, except for the CMOS-based method (i.e., method 2) and the CNTFET-based method (i.e., method 3), the initialization is necessary for the other four methods (including the proposed ternary logic), which may lead to a relatively big time delay. However, the robustness of methods 2 and 3 is not as good as the other competitors. Specifically, for methods 2 and 3, the logic operation and storage are two independent and parallel processes. That is, if the power is switched off during the logic operation, all the memory contents are erased immediately. For the other four ternary logic implementations, the logic states can be stored in the memristors (i.e., the non-volatility). Furthermore, method 2 is highly sensitive to the transistor dimensions, and method 3 suffers from the "charge pile-up" issue in the channel that may affect the performance of on/off switching. In addition, the proposed method can implement all the basic ternary logic gates, while methods 2 and 3 just provide the circuit diagrams of the ternary NOR gate and NAND gate.

## 4. Circuit Simulations and Analysis

To verify the validity and effectiveness of the entire scheme, a series of circuit simulations with the relevant analysis were conducted. The entire process description is provided below.

### 4.1. Experimental Environment

The experiment platform was a desktop workstation with a Core i7–6700 processor, 16 GB DDR4 RAM, and a Windows 10 OS. As with the other existing memristor-based logic implementation [12,16,20], the circuit experiments were also performed using PSpice (Version 16.5, Cadence, San Jose, CA, USA) and Matlab software (R2014a, MathWorks, Natick, MA, USA).

### 4.2. Parameter Selection

Based on the previous description, the parameter selection (including the device parameters and stimulation parameters) is very important for the realization of the ternary logic gates. Considering the above-mentioned constraint conditions and the device characteristics of the spintronic memristor, the parameters were chosen and are collected in Table 5.

**Table 5.** Technical parameters for the implementation of ternary logic.

| Device Parameters [1] | Stimulation Parameters | Constraint Conditions |
|---|---|---|
| $r_H = 8 \times 10^9$ Ω/m; $r_L = 4 \times 10^9$ Ω/m | $V_H = 7.0$ V | $|V_{read}| \leq R_{SH} \cdot I_{cr2}$ |
| $r_{SH} = 16 \times 10^{10}$ Ω/m; $r_{SL} = 4 \times 10^{10}$ Ω/m | $V_{Mid} = 3.0$ V | $|0.5 \cdot V_{read}| \leq R_{Mid} \cdot I_{cr2}$ |
| $J_{cr1} = 5 \times 10^{11}$; $J_{cr2} = 2.5 \times 10^{11}$; | $V_L = 0$ V | $|V_H - V_c| > R_{Mid} \cdot I_{cr2}$ |
| $H = 7 \times 10^{-9}$ m; $z = 10 \times 10^{-9}$ m | $V_C = 3.0$ V | $|V_{Mid} - V_c| \leq R_{Mid} \cdot I_{cr2}$ |
| $\Gamma_v = 1.3517 \times 10^{-11}$; $D = 10 \times 10^{-6}$ m | $V_{read} = 2.5$ V | $R_{SH} \cdot I_{cr2} \leq |V_c - V_L|$ |
| $R_{Mid} = R_{s1} = 100$ kΩ | $T = 160$ ns | $|V_{Mid}| > R_L \cdot I_{cr}$ |
| $R_{s2} = R_{s3} = 10$ Ω | / | $|V_H - V_{Mid}| > R_L \cdot I_{cr}$ |

[1] $R_L = r_L \cdot D = 4$ kΩ, $R_H = r_H \cdot D = 8$ kΩ (boundary resistance of $M_1$ and $M_2$), $R_{SL} = r_{SL} \cdot D = 40$ kΩ, $R_{SH} = r_{SH} \cdot D = 60$ kΩ (boundary resistance of $M_{s1}$), $I_{cr1} = J_{cr1} \cdot h \cdot z = 3.5 \times 10^{-5}$ A is the current threshold of the memristors $M_1$ and $M_2$, and $I_{cr2} = J_{cr2} \cdot h \cdot z = 1.75 \times 10^{-5}$ A is the current threshold of the load memristor $M_{s1}$. $V_{th1} = R_L \cdot I_{cr1} = 0.14$ V, $V_{th2} = R_H \cdot I_{cr1} = 0.28$ V (voltage thresholds of $M_1$ and $M_2$), $V_{th3} = R_{SL} \cdot I_{cr2} = 0.7$ V, $V_{th4} = R_{SH} \cdot I_{cr2} = 2.8$ V (voltage thresholds of $M_{S1}$). $V_{read}$ denotes the reading voltage, $T$ is the period of each operation.

Notably, the device parameters of the spintronic memristors (shown in Table 5) are all common values that have been proved to be valid in the literature [21]. Meanwhile, the (threshold) voltages and current densities are suitable for CMOS technology [27]. All the constraint conditions are achieved in this experiment.

### 4.3. Simulation Results and Analysis

Based on the given technical parameters, two case studies were investigated to verify the effectiveness of the proposed circuit. The specific results and the corresponding analysis are provided below.

#### 4.3.1. Case Study 1

In case study 1, the ternary NAND, NOR, AND, and OR gates were realized using the proposed hybrid memristor-CMOS logic circuit.

Figure 6 illustrates the simulation results of four basic ternary logic gates, i.e., the NAND gate (the first two rows), the NOR gate (the second two rows), the AND gate (the third two rows), and the OR gate (the last two rows). $In_1$ (the green solid line) and $In_2$ (the red dashed line) represent the input signals during the writing operation, and the purple solid line represents the obtained output logic states. Here, W(*i*, *j*) and R(*o*) denote the writing operation and reading operation, respectively. The variables *i*, *j*, and *o* represent the logic states of the input and output signals. The obtained input–output relationships were consistent with the corresponding truth tables (as shown in Tables 2 and 3), and the response time was very short (nanosecond scale). Both of these demonstrate the validity and effectiveness of the proposed memristor-CMOS based circuit diagram.

Then, the memristance variation during a ternary NAND operation is exhibited in Figure 7. The green solid line and the brown dashed line represent the resistance of memristors $M_1$ and $M_2$, respectively, and the blue solid line denotes the resistance of the load memristor $M_{s1}$. The memristance variation was in accordance with the theoretical analysis in Section 3. Meanwhile, when the power cut off, the output logic state could be stored in the load memristor $M_{s1}$ in the resistance form. Therefore, the robustness of the proposed circuit could be sufficiently guaranteed.

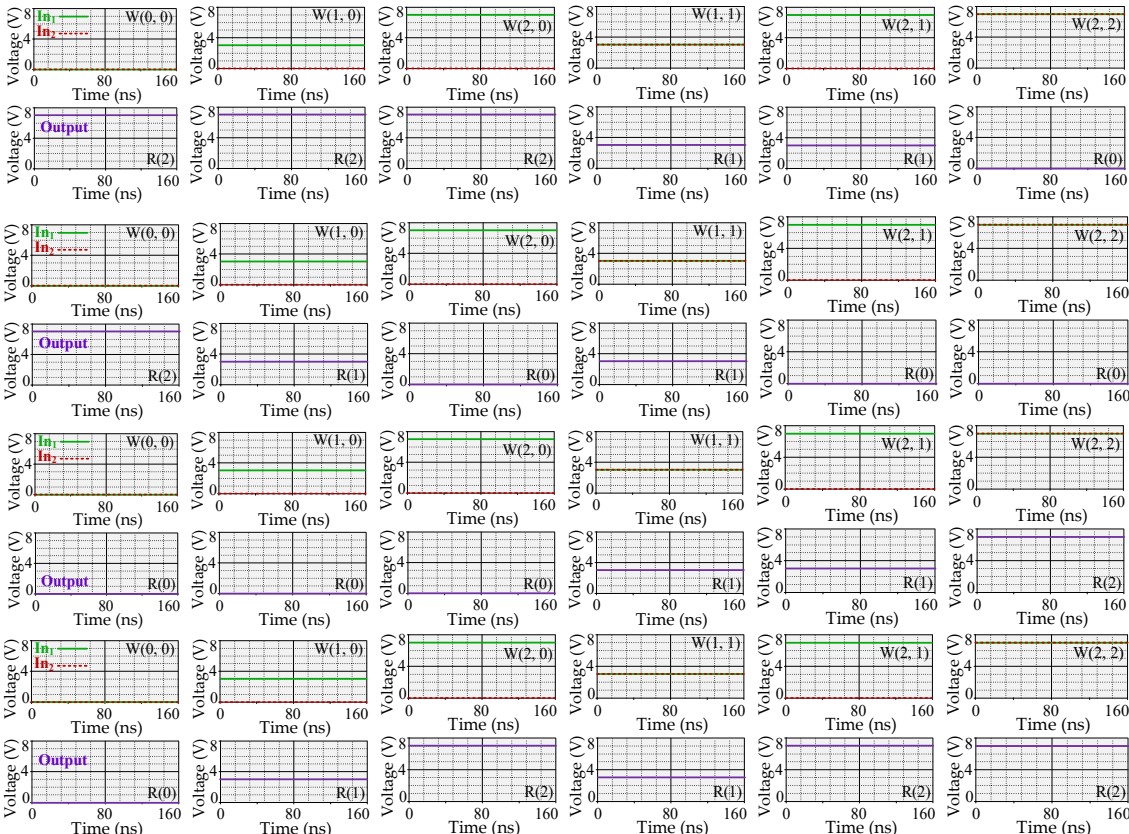

**Figure 6.** The simulation results of four basic ternary logic gates (including the NAND, NOR, AND, and OR gates).

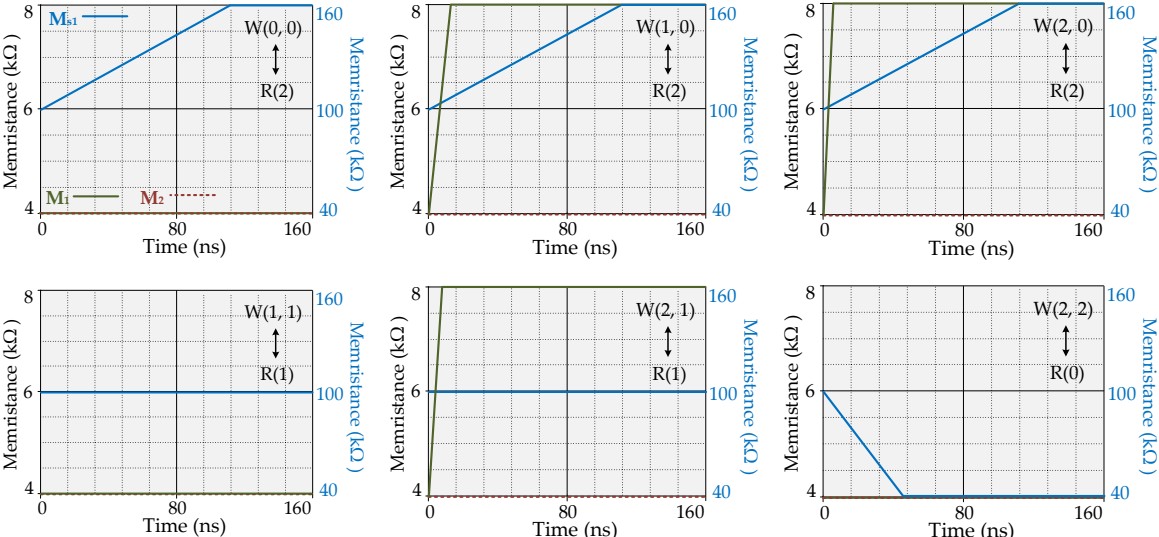

**Figure 7.** The relationship between the memristance and time during a ternary NAND operation.

### 4.3.2. Case Study 2

To demonstrate that the proposed logic circuit could perform some complicated ternary logic functions, a series of circuit simulations were conducted in this case study. The specific cascaded ternary logic function and the relevant simulation results are provided below.

Figure 8a demonstrates the specific schematic diagram and the corresponding truth table, the ternary logic function and the final circuit simulation results are provided in Figure 8b. Based on

the obtained input–output relationship, it is clear that the proposed memristor-CMOS circuit could perform this cascaded ternary logic function. In particular, the final input–output relationship was the same as that of the ternary NOR logic gate. Namely, the ternary logic function could be rewritten as:

$$\text{Function 1} = \left(\overline{\text{In}_1 + \text{In}_2}\right) \cdot \left(\overline{\text{In}_1 \cdot \text{In}_2}\right) = \left(\overline{\text{In}_1 + \text{In}_2}\right) \tag{15}$$

Notably, Equation (15) was also established for the binary logic.

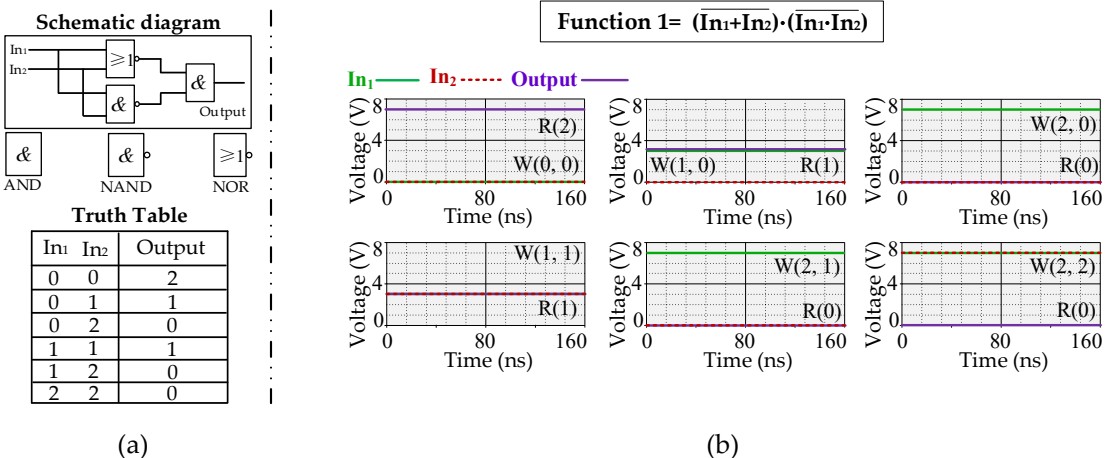

**Figure 8.** (**a**) The schematic diagram and truth table of function 1. (**b**) The circuit simulations of two cascaded ternary logic functions.

## 5. Conclusions

This study mainly investigated memristor-based unbalanced ternary logic implementation. Specifically, the spintronic memristor with its unique memristance variation rule was briefly discussed. Then, a hybrid memristor-CMOS based circuit was designed for the implementation of the basic ternary logic gates (including the NAND, NOR, AND, and OR gates). Compared with the existing ternary logic implementations, the proposed method had advantages in terms of circuit construction, response time, robustness, functionality, load capacity, and cascaded capacity. Finally, all these above-mentioned merits were verified by a series of circuit simulations.

**Author Contributions:** Methodology, H.Z.; software Z.Z. and L.L.; writing—original draft preparation, H.Z. and Z.D.; conceptualization, S.D.; writing—review and editing, H.Z. and H.L.; visualization, L.L.; supervision, Z.D.; funding acquisition, M.G. All authors have read and agreed to the published version of the manuscript.

**Funding:** This research was funded by the National Natural Science Foundation of China (grant number 61671194) and Fundamental Research Funds for the Provincial Universities of Zhejiang (grant number GK199900299012-010).

**Acknowledgments:** The authors would like to thank the editorial board and reviewers for their suggesting regarding improving this paper.

**Conflicts of Interest:** The authors declare no conflict of interest.

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
