# Peer review of "Implementation of Unbalanced Ternary Logic Gates with the Combination of Spintronic Memristor and CMOS"

_electronics, doi:10.3390/electronics9040542_

Round 1

Reviewer 1 Report

The submitted paper presents the idea that unbalanced ternary logic based on memristors can be implemented using spintronic platform. The paper presents the detailed studies of ternary logic gates implemented using spintronic memristor. The investigated memristor and gates are described and numerically investigated using SPICE model.

The presented results are interesting. The manuscript can publish in ‘Electronics’ after revision.

1. Authors should discuss more detailed the physical principles being behind the spintronic memristor. Please extend the first paragraph of Section 2.

a) Explain that the resistance dependence on the relative orientation of two coupled magnetic layers is called giant magnetoresistance and was described theoretically by Camley and Barnaś for current‑in‑plane configuration (disused in submitted manuscript)

E. Camley and J. BarnaÅ›, Theory of giant magnetoresistance effects in magnetic layered structures with antiferromagnetic coupling, Phys. Rev. Lett. 63, 664 (1989); https://doi.org/10.1103/PhysRevLett.63.664

b) Comment that the domain wall motion in a thin magnetic film driven by spin-polarized current (considered in the manuscript) described by L. Berger

L. Berger, Exchange interaction between ferromagnetic domain wall and electric current in very thin metallic films, Journal of Applied Physics 55, 1954 (1984); https://doi.org/10.1063/1.333530

Both effects (a,b) are important in electronics and spintronic nowadays. They are used to design GMR reading heads and race-track memories, and (what is crucial for submitted manuscript) both of them are combined in spintronic memristors. Authors can also cite some popular/review papers describing the perspective for nonvolatile computing or signal processing based on magnetic effects.

c) It is recommended to discuss the details of the model for spintronic memristor. Please consider citing some general paper, e.g.:

F. Nafea, A. A. S. Dessouki, S. El-Rabaie, B. E. Elnaghi, Y. Ismail, H. Mostafa, Integration An accurate model of domain-wall-based spintronic memristor, the VLSI Journal 65, 149–162 (2019), https://doi.org/10.1016/j.vlsi.2018.12.001

and/or the paper focused on CIP spintronic memristor (considered in submitted manuscript)

Y. Chen and X. Wang, Compact modeling and corner analysis of spintronic memristor, 2009 IEEE/ACM International Symposium on Nanoscale Architectures, San Francisco, CA, pp. 7-12 (2001); https://doi.org/10.1109/NANOARCH.2009.5226363

2. The choice parameters of the model listed in Table 5 should be somehow explained. The reviewer understands that the parameters are taken from Ref.[21] but some discussion should be presented.

a) Please comment if these parameters: (threshold) voltages, currents densities (in terms of energy dissipation/heat) are suitable for CMOS technology.

b) Explain if the domain wall speed (taken for the assumed current density) ensures fast enough swtiching frequency.

3. Subsection 3.3 is unclear

There are no differences between all three parts of Fig.5 because they represent the generic form of the gate. Please remove to this figure and refer to the Fig.4. The description of the AND, OR and NOR gates in Section 3.3 is unclear and must corrected.

4. The paper needs some technical corrections.

a) Captions for figures should be more descriptive.

Fig.1.  – Provide one- or two-sentence long explanation about the principle of work; mark the direction of current flow and the direction of domain wall velocity; comment that the scattering of spin-polarized current (and the resistance) depends on the size of the domains (position of domain wall).

Fig.2.  – The symbols R_H, R_L, V_th1, Vth2, and horizontal dashed lines are not explained; add (a), (b) labels in the figure.

Fig.3,4,6,7 – The captions are laconic. In Fig 6,7, please explain the meaning of the symbols: W(i,j) and R(o).

Fig.8. – Provide the mode detailed description. There is no reference to parts (a) and (b) in the caption.

b) Caption for Table 1 should explain that the code is SPICE model description for spintronic memristor.

c) The symbol D in Eq.1 and symbol M_0 are not explained. The symbol v should by subscript in Gamma_v (Eq.1-4) and in the text on page 3.

Author Response

Dear anonymous Reviewer,

Thanks very much for your constructive comments and suggestions concerning our manuscript entitled “Implementation of Unbalanced Ternary Logic gates with the Combination of Spintronic Memristor and CMOS” (No. electronics-749443). These comments and suggestions are really helpful for making this paper better, as well as promoting our research. The specific point-by-point response can be found in the uploaded attachment.

Reviewer 2 Report

The paper "Implementation of Unbalanced Ternary Logic gates with the Combination of Spintronic Memristor and CMOS" reports the realization of logic
gate with Spintronic Memristor and CMOS.
The manuscript is well written and provides a clear description of the idea, the development and the conclusion of the work.

I recommend the publication after the addressing of the following minor issues/ comments:
- in the section 2 in which you describe the memristor, I would like to recommend you to specify which kind of geometry you are considering:
current in-plane (CIP) or current perpendicular-to-plane (CPP).
- you reported in Figure 2 the possibility of accessing to multiple resistive state between Rl and Rh.
Can you comment on the maximal number of these possible states? Is this number affected or dependent over some parameters?

Author Response

(The authors gave the same response as above.)
